# Wave-driven amplification of surf-zone bottom stress on rough seabeds

Damien Sous[1], Marc Pezerat[2], Solène Dealbera[1,3,4], Héloïse Michaud[5], and Denis Morichon[1]

[1]Universite de Pau et des Pays de l'Adour, E2S UPPA, SIAME, Anglet, France
[2]SHOM Brest, France
[3]IMT Atlantique, Lab-STICC, UMR 6285, 29238, CNRS, Brest, France
[4]ODYSSEY Team-Project, INRIA Ifremer IMT-Atl., 35042, CNRS, Brest, France
[5]SHOM Toulouse, France

**Correspondence:** Damien Sous (damien.sous@univ-pau.fr)

**Abstract.** The present paper proposes a unified view of the wave-driven amplification of the wave-averaged bottom shear stress in rough seabed contexts, covering both co- and opposing wave/current cases. The analysis is first based on a series of field observations performed over the Flysch rocky shore platform of Socoa. The momentum balance is examined locally, separating the net effect of the waves on the depth- and wave-averaged momentum budget, based on velocity and pressure measurements. The present observations confirm that, in the presence of complex seabed topography, the bed shear stress is an important component of the momentum balance. The results highlight two distinct regimes depending on the breaking activity due to the intricate composition between waves and mean currents in the wave averaged shear stress. In moderately developed undertow conditions, the bottom stress brings a negative contribution to the wave momentum balance, and acts to to promote wave setdown, while in conditions of depth-limited wave breaking saturation the bed friction acts to increase the wave setup. A novel empirical parameterization of the mean bottom stress under combined waves and current is proposed. The in-situ findings are complemented by a series of wave-resolving simulations on idealized closed and open beaches, confirming the complex effect of waves on the time-averaged water circulation.

## 1 Introduction

The nearshore circulation is a key driver for the dispersion of sediments, nutrients and contaminants, controlling the renewal of coastal waters and the health of marine ecosystems. In wave-exposed areas, the transformation of short "gravity" waves in the shallowing water drives mass and momentum fluxes which affect the mean circulation and water level dynamics (Phillips, 1977; Smith, 2006) in combination with other forcings such as tides, wind stress and bottom drag.

Aiming to discriminate the wave action on nearshore waters, the wave-averaged, depth-averaged momentum equation is commonly reduced to a cross-shore balance between the barotropic pressure gradient and the divergence of the wave radiation stress tensor (Longuet-Higgins and Stewart, 1962, 1964), predicting a decrease in mean water level in the shoaling area (the wave set-down) followed by an increase in mean water level toward the shoreline once waves have broken (the wave setup). In a real context, this idealized view may lead to wave setup underestimation (Guza and Thornton, 1981; Raubenheimer et al.,

2001). In particular, in the presence of complex seabed topography, such as in rocky or coral environments, the bottom friction becomes a primary component of the momentum balance (e.g. Lowe et al., 2009; Buckley et al., 2016; Sous et al., 2020b), affecting both circulation and water level. For instance, wave setup predictions can be improved by including the bottom stress associated with the mean wave-driven offshore-directed flow (the so-called undertow) in the momentum balance (e.g. Apotsos et al., 2007). Based on laboratory experiments across a rough fringing reef, Buckley et al. (2016) identified two counteracting roughness-induced mechanisms governing the momentum balance. On the one hand, frictional wave dissipation reduces wave height, and therefore the radiation stresses prior to wave breaking and hence the wave momentum flux at breaking, resulting in a lower wave setup when compared to smooth bottom experiments. On the other hand, the action of bottom stress enhanced by roughness in the wave-averaged momentum balance increases the predictions of the wave setup for the laboratory experiments with rough configuration. This latter effect of the mean drag force on the momentum balance can be reversed in an open system with a mean current flowing with the waves, such as a barrier reef, leading in a lowering of the mean water level (Van Dongeren et al., 2013; Monismith et al., 2013; Sous et al., 2020b; Rijnsdorp et al., 2021). In addition to those two effects, the non linearity of the wave field (e.g. the wave skewness) can lead to a net contribution of the wave-averaged drag force applied by the obstacles to the mean flow, leading to a reduction of wave setup (Dean and Bender, 2006; Van Rooijen et al., 2016; Rijnsdorp et al., 2021). While these three processes are expected to act in any situation with a submerged canopy, such as rocky seabeds, coral reefs or vegetation, their relative contributions to the momentum balance and, *in fine* to the setup at the shoreline, in diverse real conditions remain to be extensively documented in the field, together with their dependency on local conditions such as roughness structure, depth or slope (Becker et al., 2014; Lavaud et al., 2022).

A primary challenge in describing the circulation over rough seabeds is to achieve an accurate representation of the mean bottom stress in the presence of waves, owing to the effect of orbital wave motions in enhancing the drag force exerted onto the mean circulation. Grant and Madsen (1979) (see also Grant and Madsen, 1986) proposed a theoretical model predicting a two-part logarithmic layer where both the wave orbital velocity and the mean current contribute to the turbulence in a wave-current boundary layer above which the turbulence scales with the mean current only, resulting in an apparent enhanced roughness for the mean flow. This model implicitly assumes the physical bottom roughness to be small in comparison with the wave current boundary layer thickness, which prevents a direct application in very rough environment (Lentz et al., 2018; Trowbridge and Lentz, 2018). Alternatively, the time average (i.e. over many wave cycles) of the instantaneous bottom stress including the combined contributions of wave and current velocities through a quadratic law could be employed in the momentum balance, including an ad-hoc bottom drag coefficient $C_d$ (Lentz et al., 2018; Buckley et al., 2016). Various approaches, often derived from steady open-channel flow dynamics, have been used to estimate $C_d$ including empirical constant value or depth-dependent formulations, either empirical (e.g. Manning-like power law) or based on the canonical turbulent boundary layer theory through the use of representative scaling of the roughness structure (Rosman and Hench, 2011; McDonald et al., 2006; Lentz et al., 2017; Boles et al., 2024). Further studies are needed to build a unified view of bottom drag formulations for rough seabeds exposed to wave action. In particular, the connection between frictional parameters (bottom drag coefficient and/or roughness length) and the architectural structure of the roughness remains sparsely documented (Rogers et al., 2018; Sous et al., 2022). More generally the validity of bottom friction formulations derived from classical steady hydraulics should be questioned

in the nearshore context where the base assumption of classical hydraulics are generally invalidated by wave action, strong unsteadiness, vertical shear, barotropic gradients and potential high roughness to depth ratio in rocky or coral areas (Chung et al., 2021).

To date, detailed examinations of friction-driven processes over rough seabeds remain sparse, in particular in the presence of waves. The present study aims to provide a novel insight of the wave-driven dynamics of rough seabed, focusing particular attention on the influence of wave orbital motion on the mean bottom stress and its impact onto the wave-averaged circulation and water level. The study is based on the combination of a dedicated field study over a rocky platform and a series of idealized simulations using phase-resolving modeling.

## 2    Methods

### 2.1    Field observations

#### 2.1.1    Study area and Instrumentation

The *Socoa* site is located near the French-Spanish border, south-west of the bay of Saint-Jean-de-Luz. Following Sous et al. (2024, 2020a), the reference seabed profile presented in Figure 1A has been reconstructed using the 10th percentile elevation over a 10 m moving window from a series of five high-resolution GNSS cross-shore profiles covering the area. The upper intertidal part of the beach, which is the area of interest for the present study, displays a gentle slope of approximately 1.5%. The beach face steepens around the spring low tide level, reaching a slope around 3%. A steeper portion is present between 12 and 22m depth before the seabed progressively flattens. The shore platform and the back cliff present the so-called *Flysch marno-calcaire de Socoa*, corresponding to a marl and limestone Flysch formation (Mulder et al., 2009; Prémaillon et al., 2021), which is representative of many rocky environments on the Basque coast. As depicted in Figure 1A,B, the Socoa platform roughness geometry presents a peculiar structure with macro-roughness, characterized by strong anisotropy with ridges oriented along the shoreline, with a typical height ranging from 0.2 to 0.8 m in the intertidal area. The ridges show a marked cross-shore asymmetry, with an upstream inclination angle between 30 and 60° and much steeper downstream faces. Following the roughness metrics analysis of Sous et al. (2024), the standard deviation and directionality index are 0.18 m and 25%, respectively.

The data analysed here have been retrieved from a larger dataset recovered during the EZPONDA campaign from September to November 2021. The present data subset was deployed on a single cross-shore transect (Figure 1) from October 6 to 18, 2021. Incoming wave conditions were provided by a Nortek Signature 1000 ® acoustic Doppler profiler (SIG) deployed at the beach toe at 20 m water depth, measuring each hour over 30 minutes burst at 4Hz. Four RBR Virtuoso ® pressure sensors, P9, P11, P12 and P13, were deployed in the intertidal zone, continuously recording bottom pressure at 5Hz. Two Nortek Vector® acoustic Doppler velocimeters were deployed at the same cross-shore location (see Figure 1D) at 0.56 (bottom ADV, named $ADV_b$) and 0.84 m (top ADV, named $ADV_t$) from the reference seabed, continuously recording velocity at 8 Hz. The bottom ADV position corresponds approximately to the top of the roughness elements.

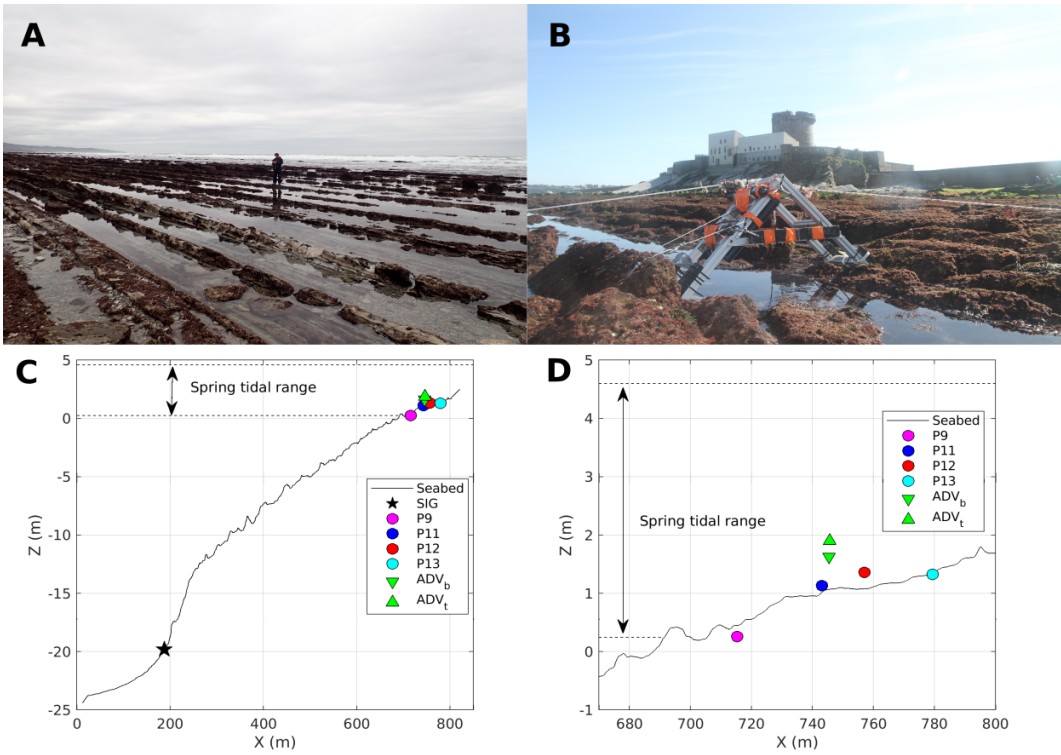

**Figure 1.** Field site and experimental setup. A: Low tide picture revealing the peculiar geometrical structure of the Socoa Flysch beachface. B: Velocity measurement structure. C: reference bathymetric profile and deployed instruments (SIG: velocity profiler, ADV: point currentmeters, P: bottom-moored pressure sensors). D: zoomed view over the control area.

### 2.1.2 Data processing

**Mean water levels and vertical positioning**

Each pressure sensor was repeatedly positioned by Real Time Kinematic Differential Global Navigation Satellite System (DGNSS-RTK). The overall uncertainty is similar to that estimated during high-resolution topography measurements performed by Sous et al. (2020a, 2024), approximately 3 and 15 cm in vertical and horizontal positioning, respectively. The mean water levels were computed assuming an hydrostatic balance from the continuous pressure records subdivided into 30-min bursts, corrected from the atmospheric pressure and sensors offset.

**Waves**

The significant wave height ($H_{m0}$) and peak period ($T_p$) of incoming waves were retrieved at SIG. The instantaneous velocity is extracted from the profile at 2.1 m above the seabed and combined with bottom pressure to reconstruct directional spectra using the Bayesian Direct Method (Hashimoto, 1997). $H_{m0}$ is computed by integrating the spectrum over the 0.05-0.35 Hz frequency band and a 270 to $10^o$ angular sector in nautical convention. For pressure sensors P9, P11, P12 and P13, free surface energy spectra were computed over 30-min bursts subdivided from the continuous record, with 10-pt spectral smoothing on

unwindowed spectral estimates (22 degrees of freedom, spectral resolution 0.005Hz). The following analysis of wave driven dynamics over the intertidal flat is focused on the data provided by P11 and P12, P9 and P13 being added only during the

105 processing to extract the incident wave parameters from the total signal (including incident and reflected waves) using the three-gauges method of Drevard et al. (2009). Incident-reflected spectral components separation is therefore performed using P9-P11-12 data for P11 and P11-P12-P13 for P12. The resulting incident bottom pressure spectra at P11 and P12 were then converted into free surface elevation spectra using linear wave theory and integrated over the 0.05-0.35 Hz frequency range to estimate the local significant wave height.

**Currents**

The wave-averaged cross-shore currents, namely $\overline{U_t}$ and $\overline{U_b}$ for top and bottom sensors respectively, are estimated by burst-averaging the instantaneous signal. A vertical profile is reconstructed using an upper parabolic profile interpolated over three points, namely the two measured velocities at their respective vertical elevations and a zero velocity at the elevation of the lowest wave crest observed during the burst, and a linear profile between the value measured at $ADV_b$ and a zero velocity

assumed at the seabed. The Eulerian cross-shore transport $Q_e$ is defined as the depth-integral of the reconstructed velocity profile. The depth-averaged, wave-averaged cross-shore velocity is then computed as:

$$U_{avg} = \frac{Q_e}{D} \tag{1}$$

where $D$ is the wave-averaged water depth.

The depth- and wave-averaged alongshore current ($V_{avg}$) is estimated as the average between the burst-averaged alongshore

currents measured at both current-meters.

### 2.1.3 Analysis framework

The dynamics is examined by considering the depth-integrated momentum budget at the measurement apparatus location, separating the net effect of the waves on the wave-averaged (Eulerian mean) momentum budget, following the approach of Smith (2006) (see also Bruneau et al., 2011).

For a steady state, with linear waves propagating toward the coast with a moderate incidence with respect to the cross-shore direction ($\theta$), and considering the alongshore uniformity and gentle slope (1.5%) of the surveyed area, the cross-shore depth- and wave-averaged momentum balance reduces to:

$$\underbrace{g\frac{\partial \eta}{\partial x}}_{M_s} = \underbrace{-\frac{\partial J_w}{\partial x}}_{M_j} + \underbrace{\frac{D_w k \cos\theta}{\sigma \rho D}}_{M_{wd}} - \underbrace{\frac{\tau}{\rho D}}_{C_d M_f} \tag{2}$$

The term of mean free surface slope $M_s$ is inferred from the gradient of the mean water level ($\eta$) between the pressure

sensors (with $g$, the gravitational acceleration).

The term $M_j$ refers to the irrotational contribution of radiation stresses, with:

$$J_w = \frac{E_w}{\rho D}\left(\frac{C_g}{C} - \frac{1}{2}\right) \tag{3}$$

where $E_w$ is the incident wave energy evaluated at pressure sensors, $C$ and $C_g$ are the wave phase and group velocities, both estimated from linear wave theory considering a wavenumber $k$ evaluated from the dispersion relation at the mean relative frequency $\sigma$, while $\rho$ is the sea water density.

$M_{wd}$ represents the dissipation of wave momentum (breaking and bottom friction), which is assumed to be transferred directly to the mean flow. The term $D_w$ is obtained from the wave action balance, that reads:

$$\frac{\partial[(C_g\cos\theta + U_{avg})A]}{\partial x} = \frac{D_w}{\sigma} \tag{4}$$

where $A = \frac{E_w}{\sigma}$ is the wave action. Finally, the friction term $C_d M_f$ is inferred from measurements at the current-meters location, using a classical quadratic law formulation of the bed shear stress $\tau$. In order to decipher the wave contribution to the mean current friction, $\tau$ is estimated either using the wave- and depth-averaged current ($\tau_{avg}$):

$$\tau_{avg} = \rho C_d \sqrt{U_{avg}^2 + V_{avg}^2}\, U_{avg} \tag{5}$$

or the full instantaneous depth-averaged Eulerian current $\tau_f$:

$$\tau_f = \rho C_d \langle \sqrt{U^2 + V^2}\, U \rangle \tag{6}$$

with $U$, $V$ the instantaneous cross-shore and alongshore velocity components averaged between both current meters, $C_d$ is a drag coefficient and the brackets denote a time average. It is worth pointing out that, contrary to the other terms based on the incident wave energy, the instantaneous velocity components encompass contributions from both incident and reflected waves. It is stressed, however, that considering irregular waves, the phase shift between these two contributions onto the orbital velocities should be randomly distributed, and hence the contributions from reflected waves presumably have no net statistical effect on the mean bed shear stress.

## 2.2 Numerical modelling

The phase-resolving wave model SWASH (Zijlema et al., 2011) is implemented along a single cross-shore transect (2DV simulations) with eight vertical layers. For each tested case, 30-min runs are performed over a 2 km domain. At the left boundary, the model is forced by a JONSWAP spectrum. A 20 m wide sponge layer is imposed at the right boundary (beach). The horizontal resolution is 1 m. A 5-min spinup period is applied before recovering the model output each 0.25 s.

All model parameters are set to default values excepted those mentioned here below. A $K - \epsilon$ turbulence model is used to provide a fine description of the vertical turbulent fluxes. At the bottom boundary, the law-of-the-wall is applied. The near-bed velocity is determined by the log-law while both $K$ and $\epsilon$ are derived from the constant bottom stress. A uniform Manning coefficient for the bottom drag is imposed with value 0.03.

Two beach geometries are tested. For both geometries, the offshore area consists in a 500m-long 30m-deep horizontal bed portion. For the first geometry, a linear slope (1:43) is imposed from X=500 m to the end of the domain at X=2000 m. This closed beach system is aimed to promote the development of undertow, corresponding to the opposite current condition. The second one is an open system, with a beach truncated by a horizontal seabed at 1m depth allowing to promote co-current

**Table 1.** Numerical test cases

| Case name | beach type | $H_s$ (m) | $T_p$ (s) |
|:---------:|:----------:|:---------:|:---------:|
| **C1** | closed | 1.5 | 7 |
| **C2** | closed | 2.5 | 10 |
| **C3** | closed | 3.5 | 13 |
| **O1** | open | 1.5 | 7 |
| **O2** | open | 2.5 | 10 |
| **O3** | open | 3.5 | 13 |

condition. Note that the flat area in the open cases does not intend to represent the field site: the numerical cases have an open boundary to allow an onshore flow while the real system is closed by a cliff. For each beach geometry, three wave forcings are tested. Test case configurations are summarized in Table 1.

The near-bed velocity is inferred from a linear interpolation of the instantaneous velocity field at 0.2m above the bottom, very similar results being obtained for other tested values ranging from 0.1 to 0.5m. The bed shear stress is then computed using Equations 5 and 6.

## 3 Results

### 3.1 Field observations

#### 3.1.1 Overview

Figure 2 depicts the wave, level and current conditions observed during the experiment. Nearly constant wave height around 1 m is observed during the first day associated with a progressive increase of the peak period. A large wave event is then observed on October 8, with peak $H_{m0}$ at 2.15 m and $T_p$ around 15 s followed by a progressive decay toward smaller and shorter wave forcing. The wave height at P11 (blue line in Figure 2A) reveals the strong tidal control over the platform, with maximum wave attenuation near low tide (Figure 2C). At high tide, wave transmission depends on the incoming wave height-to-depth ratio. For small incoming wave, the attenuation is very weak while during the large wave event, wave attenuation can reach 30% between SIG and P11 due to the combined contributions of wave breaking, bottom frictional dissipation and spectral transfers.

Figure 2D displays the burst-averaged cross-shore currents recovered at the velocity measurement station depicted in Figure 1B. One notes first that the current magnitude is strongly related to the incoming wave energy, with stronger currents observed during wave event and nearly no current for the calmer period. The cross-shore current is overwhelmingly negative, i.e. corresponding to the expected "undertow" return flow which compensates the onshore-directed mass flux in the surface layer. A straightforward tidal control is also observed: the measured cross-shore current is minimal at high tide and progressively

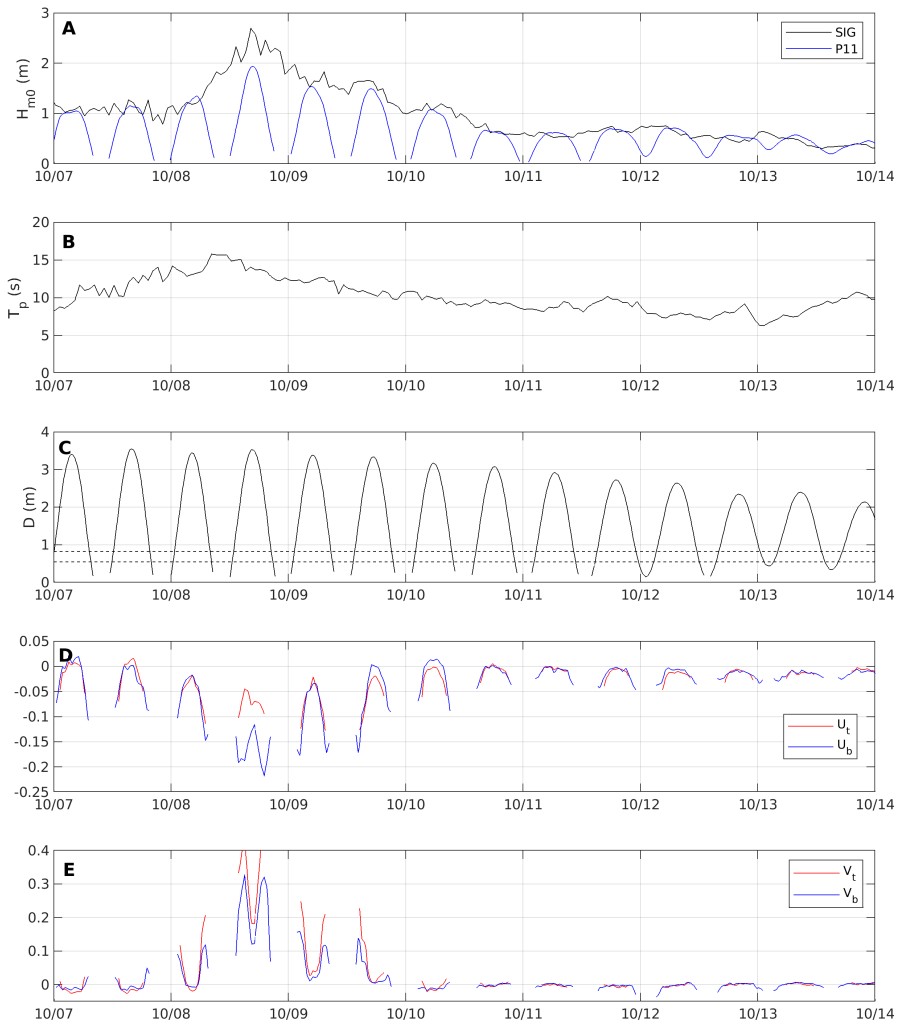

**Figure 2.** Field conditions. A: Significant wave height measured at SIG (20m depth) and P11 (intertidal platform). B: peak period at SIG. C: depth at ADV station (the dashed lines represent the ADV elevations). D: Burst-averaged cross-shore currents $\overline{U_b}$, $\overline{U_t}$ corresponding to bottom/top ADV ($ADV_b/ADV_t$), respectively. E: Burst-averaged along-shore currents $\overline{V_b}$, $\overline{V_t}$ corresponding to bottom/top ADV ($ADV_b/ADV_t$), respectively.

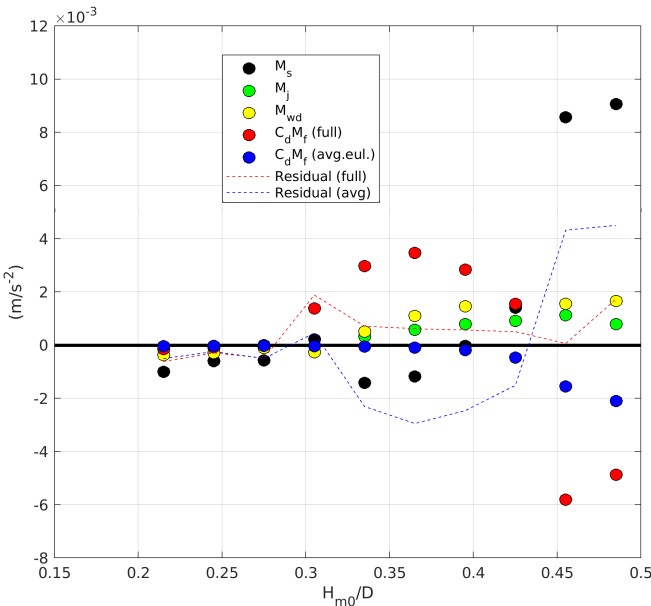

**Figure 3.** Compared momentum fluxes vs the significant wave height to local depth ratio. The colored dots represent the median contribution from each term of Eq. 2. The friction term is computed either using full instantaneous or wave-averaged velocities, in red and blue dots. The dashed lines display the related residuals for Eq. 2.

increases as depth decreases. The comparison between the two ADVs indicates an overall very good agreement, revealing a strong homogeneity of the vertical structure of time-averaged cross-shore current at the measured points. One striking exception is observed during the wave event of October, 8, where a significant vertical shear of the time-averaged current is measured. The $ADV_t$ (top) current is nearly half of the $ADV_b$ (near-bed) current, likely related to the extension of the surface onshore-directed current layer forced by the increase of incoming wave energy. Figure 2E depicts the along-shore component of velocity at both ADVs. The flow direction is mainly toward North-East. The magnitude, which increases with increasing wave height, can overcome the magnitude of the cross-shore component. However, in the cross-shore momentum analysis framework described in Section 2.1.3, the along-shore component mainly acts into the bed shear stress computation.

### 3.1.2 Momentum balance

Figure 3 displays the momentum flux dependency on the local wave height to depth ratio $H_{m0}/D$. Each colored dot displays a specific term from Eq. 2, representing the median obtained over successive $H_{m0}/D$ boxes (width 0.04). The dashed lines depict the residuals obtained for the complete balance, using the friction term either computed with the full velocity signal or the wave-averaged signals, in red and blue colors, respectively. The cross-shore momentum balance is generally dominated by the slope and the friction terms, named $M_s$ and $C_d M_f$ respectively. The bottom drag coefficient for the friction term is obtained by

minimizing the mean residual over the complete momentum balance considering the bed stress computation based on the full velocity signal. The optimized value is reached here at $C_d = 0.3$. The slope term displays a non-monotonic behavior, with a first negative peak (setdown phase) reached around $H_{m0}/D = 0.35$ and then a sign change toward positive values (setup phase) with maximum value reached for the larger $H_{m0}/D$ ratio. The friction term changes strongly depending on the estimation method. Using the wave-averaged current (blue dots in Fig. 3), the friction term remains very weak until the development of a strong undertow above $H_{m0}/D = 0.4$ which induces a negative $C_d M_f$ related to a *positive* contribution to the cross-shore momentum, i.e. to the wave setup. A drastically different response is observed using the bed stress computation based on the full velocity signal (red dots in Fig. 3). The friction term shows a first positive peak around $H_{m0}/D = 0.35$, associated with *negative* contribution to the momentum balance, before shifting to negative values for large $H_{m0}/D$ ratio, but reaching much stronger (negative) values than the friction term estimated on wave-averaged currents. The averaged residuals obtained for the full and the wave-averaged friction terms are 6.7 and $21.10^{-4}$ m/s$^{-2}$, respectively. These observations highlight the importance of the waves on the wave-averaged drag. The wave contributions, combining the irrotational $M_j$ and dissipation-driven $M_{wd}$ components, are very weak for non-breaking conditions. They both increase above between $H_{m0}/D = 0.3$ and $H_{m0}/D = 0.4$ before reaching more stable values, representing at most about 25% of the slope term.

### 3.1.3 Bed shear stress

Figure 4A and B depicts the dependency of the ratio between the full and wave-averaged velocity based stress on the ratio between the standard deviation of the velocity $U_{std}$ and $U_{avg}$. The black circles are the data points. Note that the analysis is restricted here to friction terms larger than 0.001 m/s$^{-2}$ in order to discard low signal-to-noise ratio data points.

A complex dependency is observed, highlighting the complexity of the wave-driven and wave-averaged current composition. For the classical regime of current flowing with wave ($U_{std}/U_{avg} > 0$), few data are provided by the present experiments, during very weak flow magnitude when $U_{avg} > 0$ (see Fig. 2D). Although direct interpretations should be made with caution, note that the sparse retrieved data is observed to reach much higher $U_{std}/U_{avg}$ ratio than previously documented (Lentz et al., 2018). For complementary insight, the predictions of the Wright and Thompson (1983) model are displayed in blue stars in the range of wave and current conditions where it has been typically assessed over rough seabeds (e.g. Lentz et al., 2018). More information is provided by the present dataset on undertow conditions ($U_{std}/U_{avg} < 0$), i.e. with current opposed to waves. The friction enhancement by wave is not monotonic. For moderate negative $U_{std}/U_{avg}$ ratio ($U_{std}/U_{avg} > -7$), the waves act to positively increase the friction, up to a factor about 2.5 for $U_{std}/U_{avg} = -4$. As the $U_{std}/U_{avg}$ ratio increases negatively (for instance in the presence of a relatively week undertow), the contribution of the wave orbital motion induces a sign change of the bed shear stress, which can be presumably attributed to the skewness of orbital velocities because of non-linearities in the wave field (see Appendix A). In such conditions, waves act to reverse the sign of the friction contribution to the cross-shore momentum balance.

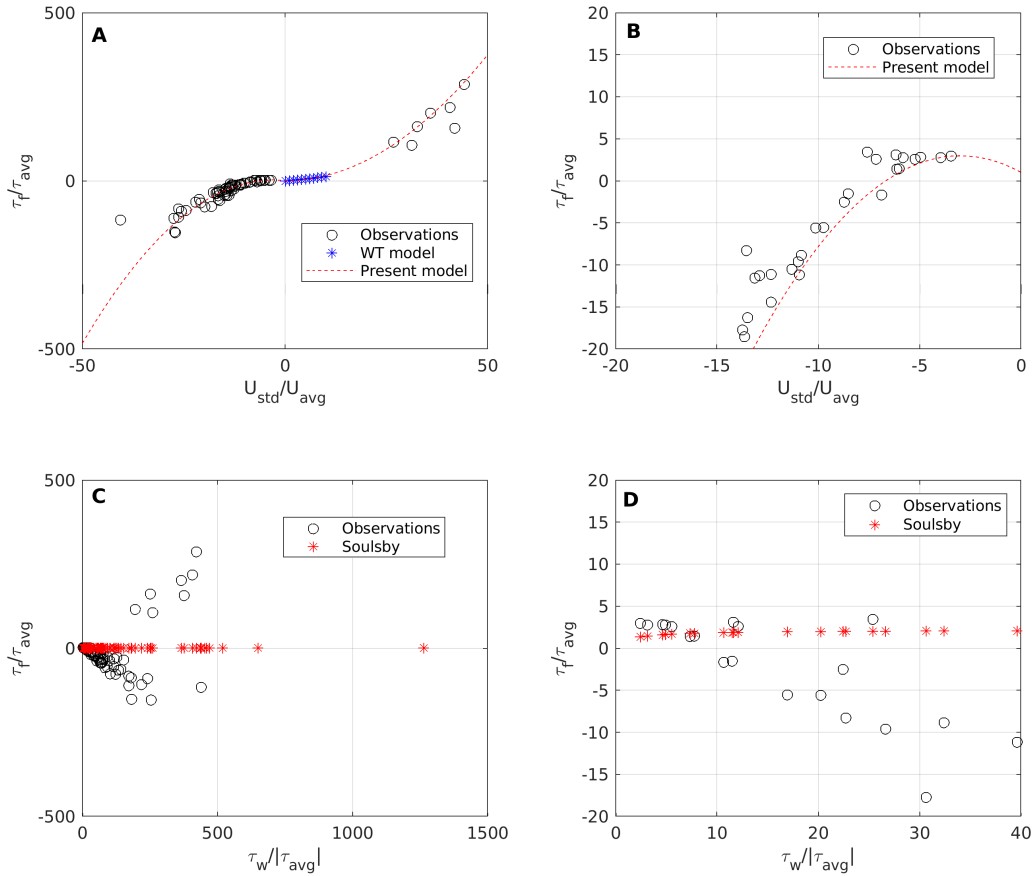

**Figure 4.** A and B (zoom of A): ratio between full shear stress based on instantaneous velocity (including wave motion) and wave-averaged shear stress. Present data are in black circle, the Wright and Thompson (1983) model in blue crosses and the proposed parameterization from Equation 7 in red line. C and D (zoom of C): comparison between the observed ratio of full vs wave-averaged shear stress vs the ratio wave-shear stress over wave-averaged shear stress and the predictions from the Soulsby's model (Soulsby, 1983).

For practical application in circulation models, an empirical parameterization is proposed based on the present dataset to estimate the complete bed shear stress from the wave-averaged shear stress:

$$\frac{\tau_f}{\tau_{avg}} = \begin{cases} 1 + 0.15\left(\frac{U_{std}}{U_{avg}}\right)^2 & \text{if } \frac{U_{std}}{U_{avg}} > 0 \\ 3 - 0.22\left(\frac{U_{std}}{U_{avg}} + 3\right)^2 & \text{if } \frac{U_{std}}{U_{avg}} < 0 \end{cases} \tag{7}$$

The parameterization shows satisfactory predictions for the overwhelming undertow (offshore) conditions observed here, but
also for onshore conditions, with few observations from the present dataset and the Wright and Thompson (1983) model. The determination coefficient between the empirical model and the present observation is 0.87. For complementary insight, Figure 4C and D display the dependency of the ratio between the full and wave-averaged velocity based stress ($\tau_f/\tau_{avg}$) with respect to the ratio between the wave-shear stress ($\tau_w$) and the magnitude of the wave-averaged stress ($|\tau_{avg}|$), comparing observations and predictions from Soulsby's model (Soulsby, 1995). This empirical parameterization, extensively used in numerical model
to account for the wave enhancement of the mean bottom stress, reads:

$$\frac{\tau_f}{\tau_{avg}} = \left(1 + 1.2\left(\frac{\tau_w}{||\tau_{avg}|| + \tau_w}\right)^{3.2}\right) \tag{8}$$

where the wave stress is classically defined from:

$$\tau_w = \frac{1}{2}\rho f_w U_w^2 \tag{9}$$

The wave friction factor $f_w$ is given by $f_w = 1.39\left(\frac{A_b}{z_0}\right)^{-0.52}$ where $A_b$ is the near-bed orbital amplitude inferred from linear
wave theory and the roughness height is estimated as $z_0 = h_r/30$, $h_r$ being four times the standard deviation of the seabed topography (Sous et al., 2023), and $U_w$ is the near-bed orbital velocity, estimated here as $U_{std}$. The Soulsby's parameterization gives an amplification of the mean bottom stress between 1.4 and 2.2 for the range of conditions considered in this study, which is consistent with the observations for weak relative contribution of the waves, i.e. the lowest $\frac{\tau_w}{|\tau_{avg}|}$ displayed in Figure 4D (corresponding to the data in region $-7 < U_{std}/U_{avg} < 0$ in Figure 4A,B). However, by definition, the Soulsby's parameteri-
zation cannot capture the sign reversal of the wave contribution in the bottom drag as the wave stress increases, which yields wrong estimates of the mean (wave-enhanced) bottom stress for $\tau_w/\tau_{avg} > 10$.

## 3.2 Numerical results

Figure 5 depicts the time-averaged parameters of the six tested cases. The closed beach cases (C1 to C3) display the expected functioning of a beach exposed to breaking waves, with the development of an undertow focused below the surf zone and
255 increasing in intensity as the wave forcing increases. The undertow is also observed over the sloping part of the open cases (O1 to O3) but an opposite shoreward flow is present on horizontal seabed area, driven by the wave-induced barotropic gradient as classically observed over open systems such as coral barrier reefs (Sous et al., 2020b). All together, these six cases allow to explore the role of waves on the time averaged bottom stress in various conditions.

Similarly to the field data presented in Figure 4, Figure 6 depicts the dependency of the ratio between the full and wave-
260 averaged velocity based stress to the ratio between the standard deviation of the fluctuating part of the velocity for the numerical

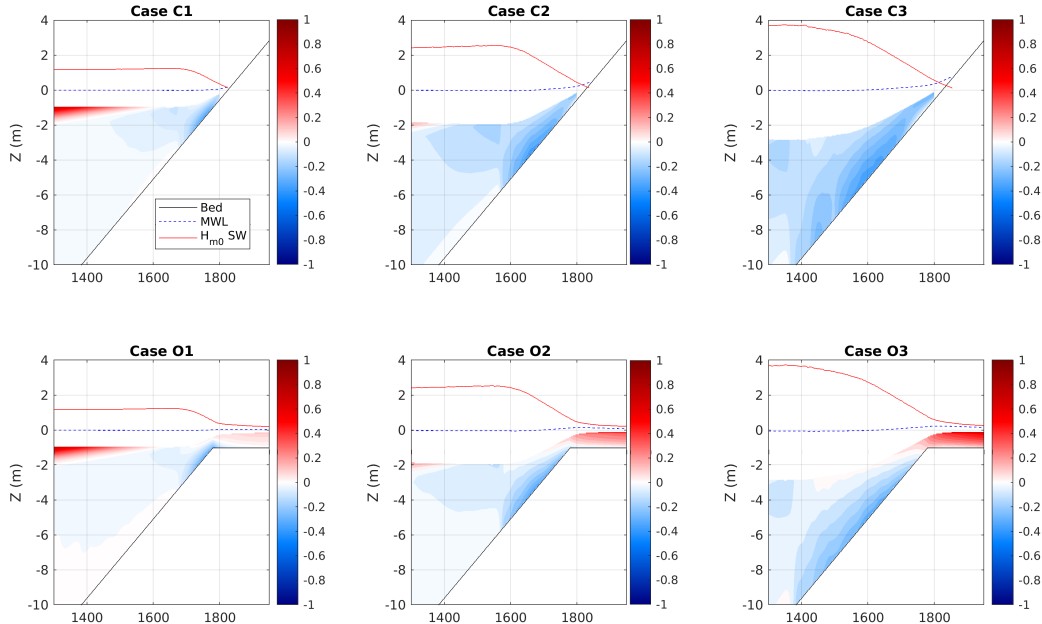

**Figure 5.** The six simulated cases. Bathymetry, significant wave height and mean water level are depicted in black, red and dashed blue lines, respectively. The color code depicts the time-averaged horizontal velocity.

simulations. Overall, the wave effect on the bottom stress shows a very similar behavior to the one revealed by the field experiments at Socoa. The numerical data is compared to the parameterization proposed in Equation 7, in red line in Figure 6. The agreement is very good for opposite wave condition (negative time-averaged current). For the positive mean near-bed current condition, the observation-based parametric model tends to underestimate the stress amplification. An adapted version

of the parameterization can be proposed to improve the agreement with the numerical data:

$$\frac{\tau_f}{\tau_{avg}} = \begin{cases} 1 + 0.3(\frac{U_{std}}{U_{avg}})^2 & \text{if } \frac{U_{std}}{U_{avg}} > 0 \\ 3 - 0.22(\frac{U_{std}}{U_{avg}} + 3)^2 & \text{if } \frac{U_{std}}{U_{avg}} < 0 \end{cases} \tag{10}$$

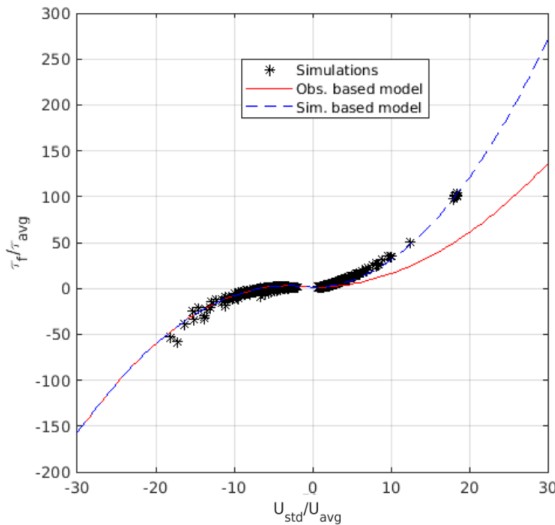

**Figure 6.** Ratio between full shear stress based on instantaneous velocity (including wave motion) and wave-averaged shear stress inferred from numerical simulations. Numerical data are in black stars while and parameterizations from Equations 7 and 10 are displayed in red and blue dashed lines for , respectively

## 4 Discussion

The present field study aims to improve our understanding of the wave-driven dynamics of rocky beaches, analyzed here in a cross-shore and depth-averaged framework. Confirming previous studies on coral, rocky or vegetated shores (Van Dongeren et al., 2013; Monismith et al., 2013; Van Rooijen et al., 2016; Sous et al., 2020b; Rijnsdorp et al., 2021), the present observations demonstrate that, in presence of strong canopy-like roughness, the bottom drag is an important component of the momentum balance. The relative effect of bed friction can reach about 70 % of the wave setup, largely dominating the irrotational contribution of radiation stresses ($M_j$) and the dissipation of wave momentum ($M_{wd}$). More specifically, the recovered data emphasize both the role played by wave action in amplifying the magnitude of the wave-averaged bed shear stress and the intricate composition between wave and mean current components in the definition of the wave-averaged shear stress direction. Two distinct regimes are observed depending on the breaking activity, providing a more complex view than the common knowledge of an unimodal action of the undertow-driven shear stress in enhancing the wave setup (e.g. Buckley et al., 2016; Lavaud et al., 2022). In the present field study, the bed friction in shoaling and weakly breaking conditions is observed to bring a negative contribution to the momentum balance. This means that, in weakly-developed undertow conditions, the bottom stress acts to increase the wave setdown. On the other hand, in conditions of depth-limited wave breaking saturation ($H_{m0}/D > 0.4$), the bed friction acts to increase the wave setup. This bimodal functioning was revealed by calculating the shear stress over the total instantaneous velocity, which provides a satisfactory residual of the wave-averaged momentum balance. Using solely the wave-averaged current in the shear stress computation leads both to a misleading interpretation of the shear stress action on the

wave setup (i.e. a systematic setup amplification) and a much stronger residual of the momentum balance, which tends to reinforce the validity of the complete computation. Following Dean and Bender (2006) or Van Rooijen et al. (2016), this behavior is interpreted as the signature of the wave-induced force applied on the roughness, inducing a reacting force applied by the roughness on the depth-averaged momentum balance. While the time-averaged effect of such force is zero for linear waves, the integration of the force applied by non-linear skewed waves, such as those surveyed here in shoaling or surf zone, over a wave cycle results in a net force. The presence of a mean current adds further complexity to the estimation of the wave-averaged force applied on the water column, with a coupled dependency to wave height, shape and current magnitude variations under the evolution of tide and wave forcing.

In the wave-averaged framework, the roughness wave-current induced force on the momentum balance can not be properly resolved at the wave scale. It is generally implemented as a wave amplification of the mean flow bottom drag coefficient. Complementary to existing studies in open conditions where the waves propagate in the same direction of mean flow (Lentz et al., 2018), or studies dedicated to the mean alongshore bottom stress (Feddersen et al., 2000), a novel empirical parameterization is proposed. It fits the classical formulation of Wright and Thompson (1983) that provides good predictive skills in such contexts, and extends the range of described regimes to opposite wave and currents and to much higher regimes in terms of ratio of orbital to mean currents as observed in the cross-shore direction of a closed, beach-like, system when an undertow starts to develop. The in-situ findings for a specific site are strengthened by the bottom stress data recovered from the idealized numerical simulations carried out using the phase-resolved model SWASH. The agreement is remarkable for the opposite current case (undertow), while further investigations should be planned to better understand the behavior for the co-current case, the present field data providing sparse observations in the co-current regime. It is worth pointing out that neither the dataset, nor the numerical simulations allow to investigate the contribution to the momentum balance nor the form of the alongshore component of the bottom stress. As such, in its current form, the empirical parameterization proposed should be restricted to the cross-shore component of the bottom stress. As the mean alongshore bottom stress can be reasonably well approached by an empirical form following Wright and Thompson (1983) as shown by Feddersen et al. (2000), a combination of both parameterizations could be presumably employed in a general 2DH setting. Accordingly, the present observations call for particular precaution when using the classical formulation of Soulsby (Soulsby, 1995) in the closed beach context.

The best-fit momentum balance is obtained with $C_d = 0.3$, which is much higher than most reported values (Rosman and Hench, 2011; Asher et al., 2016; Lentz et al., 2017) with the exception of recent extreme $C_d$ values of about 1 estimated by MacMahan et al. (2023) over much larger roughness patterns. Similar values have also been obtained over the Maupiti coral reef barrier (see Sous et al., 2022) although with much lower roughness height-to-depth ratio: the roughness standard deviation to local depth ratio is about 0.5 for Maupiti reef and ranges between 0.05 and 0.13 in the present experiments. In addition, using the $C_d$ formulation based on depth-to-maximum roughness height ratio proposed by McDonald et al. (2006) to fit their laboratory data, the present $C_d$ value would have been reached for 2 to 4 times higher roughness in the studied range of depths. Taken together, these observations tend to show a higher $C_d$ than expected based on existing knowledge. While further analysis is required to decipher the connection between frictional properties and roughness structure, a possible explanation for the increased friction at Socoa platform is the strong anisotropy of the roughness architecture, which has already been

observed to increase wave dissipation (Dealbera et al., 2024). It is worth noting that the bottom drag coefficient did not display any straightforward depth-dependency representation. This contrasts with the classical framework of steady unidirectional open channel flows where the depth-averaged bottom drag coefficient is known to increase with decreasing depth, leading to a series of empirical or semi-empirical formulations such as Manning or log-based approaches (McDonald et al., 2006; Lentz et al., 2017; Sous et al., 2022). This may be explained by the depth range studied here, limited to conditions deep enough to ensure instrument submersion, or more generally to the complexity of the present hydrodynamic context, combining in particular strong wave action, sloping bottom, depth-dependent vertical shear and strong roughness.

The present interpretations have been inferred from an idealized analysis framework, which takes into account a number of assumptions. The dynamics has been interpreted in a pure cross-shore framework. The global alongshore uniformity and gentle slope of the study zone tends to assume minimal transverse effects in the balance, but more comprehensive instrumentation should be deployed to achieve a full control of advective and vortex force terms. The presented momentum balance discards the roller effect. Alternative formulations including roller have been tested (Bruneau et al., 2011) based on parameterizations of roller features (Dally and Brown, 1995; Martins et al., 2018; Streßer et al., 2022) and assuming further strong hypothesis. While slightly different values have been obtained, the roller-including formulations neither change the overall orders of magnitude and proposed interpretations nor improve the momentum balance residual, which leads us to favor the presentation of the simpler form. Similarly, a more traditional framework based on the total momentum budget (Phillips, 1977) has been tested (Gourlay and Colleter, 2005; Buckley et al., 2015; Monismith et al., 2024), showing slightly degraded momentum residuals compared to the present approach but leading to the same conclusions. Waves have also been assumed to be linear, which is certainly a rough approximation of surf zone waves. However, while accurate approaches have been proposed to non-linearly reconstruct the free surface elevation time-series from bottom pressure measurements (Bonneton et al., 2018; Martins et al., 2021), they are not compatible with the spectral incident/reflected separation performed here, which remains an important driver of the energy flux computation. The reflection coefficient, estimated in the sea swell band using the three-gauge method at P9-P11-P13, varies from 0.05 to 0.15 during the monitored period. Neglecting the effect of reflection, i.e. using the full wave signal for the momentum balance assessment, affects the estimation of the wave momentum dissipation term ($M_{wd}$ in Equation 2) due to the inclusion of reflection-based spatial modulation patterns in the gradient of wave action (Eq. 4). This degrades the overall residual, which justifies here the use of separation, but does not affect the general conclusion reached here. Assumptions have also been done on the shape of the vertical velocity profile, which will require extensive validation by further detailed experiments.

## 5 Conclusions

Aiming to improve our knowledge of the dynamics of nearshore waters in rough environments, the present study focuses on the bottom drag in the presence of waves. The overall analysis framework is the time-averaged and depth-averaged momentum balance. The study combines dedicated measurements on a rocky platform and a series of phase-resolving simulations over idealized closed and open beach profiles. The main field-based findings are (i) the key role played by bottom drag in

the momentum balance, (ii) the importance of considering the complete instantaneous near-bed velocity (i.e. not only the wave-averaged current) in the bottom stress estimation, (iii) the complex combination of orbital and mean current velocity in determining the wave-averaged shear stress and (iv), the absence of straightforward depth-dependency of the bottom drag coefficient in the studied conditions. We propose a novel parameterization to predict the wave-driven amplification of the bottom stress in both opposite and co-current regime. Further research works should be engaged to assess the validity of the present findings in a wider range of contexts.

*Data availability.* The presented data can be freely available at https://doi.org/10.17882/105486

*Author contributions.* DS, SD and DM designed the experiments. DS, SD, HM and DM carried them out. DS processed the field data and performed the simulations. DS and MP performed the formal analysis. DS and MP prepared the manuscript with contributions from all co-authors.

*Competing interests.* No competing interests are present

*Acknowledgements.* This study was carried out as part of the project PROTEVS2 under the auspices of French Ministry of Defense / DGA, and led by Shom and Université de Pau et des Pays de l'Adour. The UPPATECH SCOPE platform and the GLADYS group provided the instrumentation.

## Appendix A: Decomposition of the cross-shore bottom shear stress under the weak-current approximation

We propose here a formal analysis of the cross-shore bottom stress decomposition. While not allowing a conclusive analytical formulation at this stage, this informational approach may be used as a base for future developments. Following Feddersen et al. (2000), we assume $U$, $V$ are joint-Gaussian distributed random variables written as mean and fluctuating components, such that $[U, V] = [U_{avg}, V_{avg}] + [U', V']$, with variances $\sigma_U^2$ and $\sigma_V^2$, respectively; the cross-shore component of the bed shear stress can be expressed as:

$$\frac{\tau_f}{\rho C_d} = \langle \sqrt{\left(x + \frac{U_{avg}}{\sigma_T}\right)^2 + \left(y + \frac{V_{avg}}{\sigma_T}\right)^2} \left(x \frac{\sigma_T}{U_{avg}} + 1\right)\rangle \sigma_T U_{avg} \tag{A.1}$$

where $\sigma_T^2 = \sigma_U^2 + \sigma_V^2$, $x = \frac{U'}{\sigma_T}$ and $y = \frac{V'}{\sigma_T}$ and the brackets formally denotes the expectation operator. Assuming weak-currents (i.e. small $\frac{U_{avg}}{\sigma_T}$ and $\frac{V_{avg}}{\sigma_T}$), a Taylor expansion of the square root keeping up to linear terms in the mean current, yields:

$$\frac{\tau_f}{\rho C_d} = \underbrace{\langle \sqrt{x^2+y^2}\left(1+\frac{x}{x^2+y^2}\frac{U_{avg}}{\sigma_T} + \frac{y}{x^2+y^2}\frac{V_{avg}}{\sigma_T}\right)\left(x\frac{\sigma_T}{U_{avg}}+1\right)\rangle}_{\mathcal{E}}\sigma_T U_{avg} \tag{A.2}$$

The expectation being a linear operator, $\mathcal{E}$ can be decomposed into:

$$\mathcal{E} = \langle x\sqrt{x^2+y^2}\rangle\frac{\sigma_T}{U_{avg}} + \langle\sqrt{x^2+y^2}\rangle + \langle\frac{x}{\sqrt{x^2+y^2}}\rangle\frac{U_{avg}}{\sigma_T} + \langle\frac{x^2}{\sqrt{x^2+y^2}}\rangle + \langle\frac{y}{\sqrt{x^2+y^2}}\rangle\frac{V_{avg}}{\sigma_T} + \langle\frac{xy}{\sqrt{x^2+y^2}}\rangle\frac{V_{avg}}{U_{avg}} \tag{A.3}$$

Considering that the underlying joint probability density function is purely gaussian:

$$\langle x\sqrt{x^2+y^2}\rangle = \langle\frac{x}{\sqrt{x^2+y^2}}\rangle = \langle\frac{y}{\sqrt{x^2+y^2}}\rangle = 0 \tag{A.4}$$

As a result, one can obtain the following:

$$\mathcal{E} = \langle\frac{2x^2+y^2}{\sqrt{x^2+y^2}}\rangle + \langle\frac{xy}{\sqrt{x^2+y^2}}\rangle\frac{V_{avg}}{U_{avg}} \tag{A.5}$$

Note that the weak-current form of the Wright and Thompson (1983) parameterization is recovered assuming an isotropic wave field (i.e. $\sigma_U = \sigma_V$) and neglecting the mean longshore velocity, yielding (Feddersen et al., 2000):

$$\frac{\tau_f}{\rho C_d} = \frac{3\sqrt{\pi}}{4}\sigma_T U_{avg} \tag{A.6}$$

It is worth pointing out that in the presence of non-linearities in the wave field, the velocity skewness is usually non-zero such that $\langle x\sqrt{x^2+y^2}\rangle \neq 0$. Consequently, as $\frac{U_{avg}}{\sigma_T} \to 0$, the first term on the right-hand side of Eq. A.3 ($\langle x\sqrt{x^2+y^2}\rangle\frac{\sigma_T}{U_{avg}}$) becomes dominant. However, the analytical evaluation of the resulting bed shear stress is not straightforward, which therefore motivates the use of empirical parameterization.

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
