# Peer review of "Wave-driven amplification of surf-zone bottom stress on rough seabeds"

_EGUsphere, 2025_

## Referee Comment (RC1)

**Review of manuscript – egusphere-2025-2285**

**title**: Wave-driven amplification of surf-zone bottom stress on rough seabeds.

**authors**: Damien Sous, Marc Pezerat, Solène Dealbera, Héloïse Michaud, and Denis Morichon.

**1 Major comments**

**1.1 Alongshore velocity $V$**

The paper does not present any information on the alongshore component $V$ of the velocity (magnitude, direction or variability), and how it compares with the cross-shore component $U$. Is the magnitude of $V$ just very small compared to the magnitude of $U$?

Since the alongshore direction is part of equations (5) and (6), which are essential to the paper, $V$ must be discussed in the paper. Furthermore, I am worried that the ADVs do not give accurate $V$ because of flow distortion by the frame and alignment of the ADV's bodies in the alongshore direction.

**1.2 Reflection**

Wave reflection is brought up a few times (lines 97-98, 140-142, and 294), and motivates computing wave statistics based on the directional wave spectrum. However, there is no mention of how large wave reflection is at this site. How large is reflection? Have you also computed the momentum balance from wave statistics based on frequency(-only) spectra and do the results look significantly different? Could you recommend to the reader whether computing wave statistics based on the signal of incident waves alone is essential to get results you got?

**1.3 Definition of near-bed velocity data point**

I don't understand the velocity data point in (i), lines 104-106. There might be a more comprehensive explanation in Sous et al. (2024), but the current manuscript should include a clearer explanation of data point (i).

1. Data point is the one closest to the seabed, but it is not clear what height that corresponds to. What is the height of "two times the standard deviation of the fine seabed topography"?

2. What is the reasoning for placing the average between $\overline{U_t}$ and $\overline{U_b}$ *below* the heights of the bottom-most measurement instead of the average height?

3. If you are taking an average, shouldn't data point (i) be irrelevant for the parabolic fit because the average is a linear combination of $\overline{U_t}$ and $\overline{U_b}$?

**2  Minor comments**

1. Numerical simulation: Please indicate in Section 2.2 and/or 3.2 the correspondence between the field site and the geometry of the domain in the numerical simulations. Since your field site is described as a rocky platform, it appears that the flat region in numerical simulations O1-O3 would represent your field site (which is not the case).

2. l. 5 and 23: Except for these instances, the word "roughness" refers to the shape of the seabed. Because the word refers to a geometrical property, "well-developed seabed roughness" is confusing.

3. l. 9-10: You contrasted "lower mean water level" with "setup". I suggest rewriting to either contrast setdown with setup, or lower mean water level with higher mean water level.

4. l. 9 and 248: Here you wrote "while under saturation breaking conditions". This wording is a bit confusing because "undersaturation" is a scientific term in itself, I suggest rewriting, where you could use "in conditions of depth-limited wave breaking saturation".

5. l. 31: Rewrite "rough experiments". Perhaps laboratory experiments simulating rough seabeds?

6. l. 35: The word "water" here is unnecessary.

7. l. 73: Word "with" appears twice.

8. l. 79: It is stated that 4 pressure sensors were deployed, and the identification numbers are P9 to P13, which implies that 5 sensors were deployed. Please rephrase the sentence or change instrument identification to avoid confusion.

9. l. 92-93: Could you add a couple .

10. l. 173: Based on Fig. 2, isn't it better to say wave event in the singular?

11. l. 179: The terminology in parenthesis is swapped between $ADV_t$ and $ADV_b$.

12. Section 3.1.2: When referencing the dots in Fig. 3, specially when differentiating between the frictional terms, I suggest referencing the color of the dots in parenthesis after referencing te corresponding term – e.g., in l. 191, write Using the wave-averaged current (blue dots in Fig. 3).

13. l. 209: commas misplaced and confusing

14. l. 210: typo. Should be "where it has been".

15. l. 256: Missing parenthesis around the Feddersen et al. reference.

16. Fig. 3: To improve the figure, add a horizontal line in the background along the coordinate 0 in the ordinate.

17. Figs. 4 and 6: Replace symbols $U'$ and $U_{\mathrm{m}}$ used in these figures to match the body of the manuscript ($U_{std}$ and $U_{avg}$).

18. Fig. 5: I recommend using a divergent red-blue colormap. Replace Hs with $H_{m0}$ because you use the latter throughout the paper.

19. Section 3.1.3: If I understood correctly, $U_{avg} > 0$ can only happen at times when at least either $\overline{U_t}$ or $\overline{U_b}$ are greater than 0 in Fig. 2D. From this figure, this only happens for tiny magnitude of $U_{avg}$. Therefore, it seems the errors for the corresponding data points in Fig. 4A (i.e., in the onshore regime) should be very big. Remarkably, Figs. 4A and 4C show the results from the onshore case are reasonably consistent with expectations. Please either remove these data points if you think they are not reliable, or add a word of caution regarding the fit to these observations at $U_{avg} > 0$.

---

## Author Comment (AC1)

**Response to reviewer**

September 26, 2025

We would like first to thank the reviewer for his/her careful reading and constructive criticism. The suggested modifications have been a decisive help to revise and improve our manuscript. Specific comments are addressed one by one hereafter. Reviewers comments are recalled below in bold font, while text extractions from the revised paper are in italic font.

**1 Major Comments**

**1.1 Along-shore velocity**

**The paper does not present any information on the alongshore component V of the velocity (magnitude, direction or variability), and how it compares with the cross-shore component U . Is the magnitude of V just very small compared to the magnitude of U ? Since the alongshore direction is part of equations (5) and (6), which are essential to the paper, V must be discussed in the paper. Furthermore, I am worried that the ADVs do not give accurate V because of flow distortion by the frame and alignment of the ADV's bodies in the alongshore direction.**

Timeseries of alongshore current have been added in Figure 2 (see panel E). The flow is mainly toward North-West, so the ADVs are fully exposed to the alongshore drift. The magnitude of the along-shore component is not weak, it can even be stronger than the cross-shore component. However, under the assumption of a along-shore uniform configuration, the along-shore component contribution to the cross-shore momentum balance should be limited to the shear stress computation. The following details have been included at the end of Section 3.1.1.

*Figure 2E depicts the along-shore component of velocity at both ADVs. The flow direction is mainly toward North-East. The magnitude, which increases with increasing wave height, can overcome the magnitude of the cross-shore component. However, in the cross-shore momentum analysis framework described in Section 2.1.3, the along-shore component mainly acts into the bed shear stress computation.*

**1.2 Reflection**

**Wave reflection is brought up a few times (lines 97-98, 140-142, and 294), and motivates computing wave statistics based on the directional wave spectrum. However, there is no mention of how large wave reflection is at this site. How large is reflection? Have you also computed the momentum balance from wave statistics based on frequency(- only) spectra and do the results look significantly different? Could you recommend to the reader whether computing wave statistics based on the signal of incident waves alone is essential to get results you got?**

The following details have been added at the end of the Discussion section:

*The reflection coefficient, estimated in the sea swell band using the three-gauge method at P9-P11-P13, varies from 0.05 to 0.15 during the monitored period. Neglecting the effect of reflection, i.e. using the full wave signal for the momentum balance assessment, affects the estimation of the wave momentum dissipation term ($M_{wd}$ in Equation 2) due to the inclusion of reflection-based spatial modulation patterns in the gradient of wave action (Eq. 4). This degrades the overall residual, which justifies here the use of separation, but does not affect the general conclusion reached here.*

The momentum balance plot with full signal is displayed here in Figure 1.

[Figure]

Figure 1: Compared momentum fluxes based on the full wave energy spectra

**1.3 Definition of near-bed velocity data point**

**I don't understand the velocity data point in (i), lines 104-106. There might be a more comprehensive explanation in Sous et al. (2024), but the current manuscript should include a clearer explanation of data point (i).**

1. **Data point is the one closest to the seabed, but it is not clear what height that corresponds to. What is the height of "two times the standard deviation of the fine seabed topography"?**

2. **What is the reasoning for placing the average between Ut and Ub below the heights of the bottom-most measurement instead of the average height?**

3. **If you are taking an average, shouldn't data point (i) be irrelevant for the parabolic fit because the average is a linear combination of Ut and Ub ?**

We do agree with the reviewer, this was unclear. A simpler reconstruction is proposed, described in Section 2.1.2 (§ Currents) in the revised version as follows:

*A vertical profile is reconstructed using an upper parabolic profile interpolated over three points, namely the two measured velocities at their respective vertical elevations and a zero velocity at the elevation of the lowest wave crest observed during the burst, and a linear profile between the value measured at $ADV_b$ and a zero velocity assumed at the seabed.*

[Figure]

Figure 2: Example of reconstructed profile

An illustration of a reconstructed profile is depicted in Figure 2 here.

It should be stressed that such reconstruction is motivated by the fact that the depth-averaged framework adopted here assumes a depth-uniform wave-averaged velocity, whereas the flow actually depicts a strong vertical shear. Computing $U_{avg}$ from a simple average of $\overline{U_t}$ and $\overline{U_b}$ would result in an overestimated depth-averaged Eulerian cross-shore transport.

**2 Minor comments**

1. **Numerical simulation: Please indicate in Section 2.2 and/or 3.2 the correspondence between the field site and the geometry of the domain in the numerical simulations. Since your field site is described as a rocky platform, it appears that the flat region in numerical simulations O1-O3 would represent your field site (which is not the case).**

   The following precision has been added in Section 2.2:
   *Note that the flat area in the open cases does not intend to represent the field site: the numerical cases have an open boundary to allow an onshore flow while the real system is closed by a cliff.*

2. **l. 5 and 23: Except for these instances, the word "roughness" refers to the shape of the seabed. Because the word refers to a geometrical property, "well-developed seabed roughness" is confusing.**

   The two instances have been modified to *complex seabed topography*

3. **l. 9-10: You contrasted "lower mean water level" with "setup". I suggest rewriting to either contrast setdown with setup, or lower mean water level with higher mean water level.**

   The sentence has been rewritten in terms of setup/setdown only.

4. **l. 9 and 248: Here you wrote "while under saturation breaking conditions". This wording is a bit confusing because "undersaturation" is a scientific term in itself, I suggest rewriting, where you could use "in conditions of depth-limited wave breaking saturation".**

Both sentences have been corrected following the reviewer's suggestion.

5. **l. 31: Rewrite "rough experiments". Perhaps laboratory experiments simulating rough seabeds?**

"Rough experiments" has been replaced by *the laboratory experiments with rough configuration*

6. **l. 35: The word "water" here is unnecessary.**

Word removed

7. **l. 73: Word "with" appears twice.**

Correction applied

8. **l. 79: It is stated that 4 pressure sensors were deployed, and the identification numbers are P9 to P13, which implies that 5 sensors were deployed. Please rephrase the sentence or change instrument identification to avoid confusion.**

It is indeed confusing. It is now precised that the four pressure sensors are P9, P11, P12 and P13

9. **l. 92-93: Could you add a couple .**

The following details have been added:
*The instantaneous velocity is extracted from the profile at 2.1 m above the seabed and combined with bottom pressure to reconstruct directional spectra using the Bayesian Direct Method (Hashimoto, 1997). $H_{m0}$ is computed by integrating the spectrum over the 0.05-0.35 Hz frequency band and a 270 to 10° angular sector.*

10. **l. 173: Based on Fig. 2, isn't it better to say wave event in the singular?**

Yes, correction done.

11. **l. 179: The terminology in parenthesis is swapped between ADVt and ADVb .**

Thank you for pointing it out, correction done.

12. **Section 3.1.2: When referencing the dots in Fig. 3, specially when differentiating between the frictional terms, I suggest referencing the color of the dots in parenthesis after referencing te corresponding term – e.g., in l. 191, write Using the wave-averaged current (blue dots in Fig. 3).**

Thanks for this good suggestion. Correction applied.

13. **l. 209: commas misplaced and confusing**

   Right, commas removed.

14. **l. 210: typo. Should be "where it has been".**

   Correction done

15. **l. 256: Missing parenthesis around the Feddersen et al. reference.**

   Correction done

16. **Fig. 3: To improve the figure, add a horizontal line in the background along the coordinate 0 in the ordinate.**

   Correction done

17. **Figs. 4 and 6: Replace symbols U 0 and Um used in these figures to match the body of the manuscript (U std and Uavg ).**

   Correction done

18. **Fig. 5: I recommend using a divergent red-blue colormap. Replace Hs with Hm0 because you use the latter throughout the paper.**

   Correction done

19. **Section 3.1.3: If I understood correctly, Uavg > 0 can only happen at times when at least either Ut or Ub are greater than 0 in Fig. 2D. From this figure, this only happens for tiny magnitude of Uavg . Therefore, it seems the errors for the corresponding data points in Fig. 4A (i.e., in the onshore regime) should be very big. Remarkably, Figs. 4A and 4C show the results from the onshore case are reasonably consistent with expectations. Please either remove these data points if you think they are not reliable, or add a word of caution regarding the fit to these observations at Uavg > 0.**

   That is right, the following precision have been added in section 3.1.3:
   *For the classical regime of current flowing with wave ($U_{std}/U_{avg} > 0$), few data are provided by the present experiments, during very weak flow magnitude when $U_{avg} > 0$ (see Fig. 2D). Although direct interpretations should be made with caution, note that the sparse retrieved data is observed to reach much higher $U_{std}/U_{avg}$ ratio than previously documented (Lentz et al., 2018).*

---

## Author Comment (AC2)

**Response to reviewer**

September 26, 2025

We would like first to thank Dano Roelvink for his careful reading and constructive criticism. The suggested modifications have been decisive help to revise and improve our manuscript. Specific comments are addressed one by one hereafter. The comments are recalled below in bold, while text extractions from the revised paper are in italic font.

**1 Major Comments**

1. **In the introduction, the authors refer to some, but only very limited, research on coral reefs, which have similarly high roughness; e.g. in van Dongeren et al., (2013), where the roughness, with similar values, was found to dominate the energy balance. But similarly, much work has been done on the effects of vegetation, which has similar effects as high roughness; van Rooijen et al (2013) for instance elucidate the mechanism through which vegetation can lead to lowering of the setup, namely through streaming and skewness of the waves, which leads to an onshore force on the vegetation (and similarly on the rocky bottom in this case). In the discussion of the resulting empirical relationship I miss this kind of analysis**

Additional elements, including physical interpretation and associated bibliographical references, have been added in the Introduction section (see last part of the second paragraph) and in the Discussion section (first paragraph).

2. **The authors refer to Feddersen et al. (2000) for the mean longshore shear stress due to current and waves, but only very briefly. The cited work contains an in-depth analysis of possible approximations of the resulting mean shear stress, and based on their data select and fit one. I can see that the cross-shore mean shear stress is more complicated because of the effects of skewness but one could at least try better to explain the relationships that are found.**

Thank you for pointing this out. The physical interpretation of the time-averaged shear stress is quite challenging for the cross-shore setting considered in this study, and deserves further discussion. When considering the alongshore bottom stress parameterizations as discussed by Feddersen et al. (2000) and former pioneering studies (e.g. Longuet-Higgins, 1970, Thornton and Guza, 1986), the wave direction has no net impact onto the direction of the mean alongshore stress that is conformal with the mean alongshore current. The magnitude of the mean alongshore bottom stress is further shown to depend on both the mean alongshore current and the full statistics of orbital motion, and can be reasonably well approach by an empirical form following Wright and Thompson (1983) as shown by Feddersen et al. (2000) (see their Fig. 8). For the cross-shore component of the mean bottom stress, the wave direction is closely aligned with

the mean cross-shore current, as a result, the direction (i.e. the sign) of the mean bottom stress can change depending on vector composition. The resulting stress depends on the mean cross-shore current and the statistics of the orbital motion in a complex non-linear fashion. The wave skewness should a crucial role, and we take advantage of the reviewer comment to propose an informational formal analysis of the bed shear stress term following Feddersen et al. (2000) (see the new Appendix A).

3. **To support the claim that the relationship(s) for the cross-shore bed shear stress can be used in circulation models, we would have to know how in a general 2DH setting the cross-shore and longshore shear stresses should be computed, somehow combining these relationships with Feddersen's model (?)**

and

4. **Perhaps the parametric relationships by Soulsby et al. could be useful in this context, as they give relationships for taum/(tauc+tauw) as a function of tauc/(tauc+tauw), where taum is the mean shear stress, tauc the current-related and tauw the wave-related shear stress. The parametric relations are fitted to various wave-current interaction models, for arbitrary magnitudes and angles between current and waves. But maybe the authors have a better idea how to approach my point 3.**

Both comments are addressed hereafter. The present study provides insights on the particular case of waves propagating against the mean flow, as observed in the cross-shore direction of a closed, beach-like, system when an undertow starts to develop, and is further complemented with idealized numerical simulations of both open and closed systems. Neither the dataset nor the numerical simulations allow to investigate the contribution to the momentum balance nor the form of the alongshore component of the bottom stress. As such, in its current form, the empirical parameterization proposed should be restricted to the cross-shore component of the bottom stress. As the mean alongshore bottom stress can be reasonably well approached by an empirical form following Wright and Thompson (1983) as shown by Feddersen et al. (2000), a combination of both parameterizations could be presumably employed in a general 2DH setting. Classically used, the parametric relationships by Soulsby (1995) present the practical advantage of providing a similar form for both components of the wave enhanced bottom shear stress. For complementary insight, two subplots have been added in the Figure 4 of the revised paper (reproduced below) to display the ratio between the full and wave-averaged velocity based stress ($\tau_f/\tau_{avg}$) with respect to the ratio between the wave-shear stress ($\tau_w$) and the magnitude of the wave-averaged stress ($\tau_w/|\tau_{avg}|$), comparing observations and predictions from Soulsby's model. The Soulsby's parameterization gives an amplification of the mean bottom stress between 1.4 and 2.2 for the range of conditions considered in this study, which is consistent with the observations for weak relative contribution of the waves, i.e. the lowest $\frac{\tau_w}{|\tau_{avg}|}$ values. However, by definition, the Soulsby's parameterization cannot capture the sign reversal of the wave contribution in the bottom drag as the wave stress increases, which yields wrong estimates of the mean (wave-enhanced) bottom stress for $\tau_w/\tau_{avg} > 10$.

The relative performance of Soulsby's parameterization is presented in Section 3.1.3 of the revised version of the paper, while the practical relevance of the proposed parameterization is discussed in Section 4 (second paragraph).

**2  Minor comments**

1. **l. 190 depend $\longrightarrow$ depending**

[Figure]

Figure 1: A and B (zoom of A): ratio between full shear stress based on instantaneous velocity (including wave motion) and wave-averaged shear stress. Present data are in black circle, the Wright and Thompson (1983) model in blue crosses and the proposed parameterization from Equation **??** in red line. C and D (zoom of C): comparison between the observed ratio of full vs wave-averaged shear stress vs the ratio wave-shear stress over wave-averaged shear stress and the predictions from the Soulsby's model (Soulsby, 1995).

Done

**2. l. 277 For Chezy, drag coefficient does not depend on depth ($C_d = g/C^2$)**

Right, Chezy has been removed from the list.

---

## Author Response (AR2)

**Response to editor**

November 3, 2025

Dear Anne-Marie

The remark from Reviewer1 about current direction has been taken into consideration (the direction reads now North-East).

About the Reviewer2 comment on the relevancy of plotting the Soulby's formulation, even if it fails to predict the observed stress enhancement, we think it deserves to be displayed owing to the fact that it is a very classical parameterization used in many models. But if it is considered more appropriate by the Reviewer and the Editor, the plots can be easily moved to appendix.

Best regards,

Damien Sous, on the behalf of all co-authors